# Plasma Interleukin-10 and Cholesterol Levels May Inform about Interdependences between Fitness and Fatness in Healthy Individuals

**DOI:** 10.3390/ijerph18041800

**Published:** 2021-02-12

**Authors:** Francesco Sartor, Jonathan P. Moore, Hans-Peter Kubis

**Affiliations:** 1Department of Patient Care and Monitoring, Philips Research, 5656 AE Eindhoven, The Netherlands; 2College of Human Sciences, Bangor University, Bangor LL57 2EF, UK; j.p.moore@bangor.ac.uk (J.P.M.); h.kubis@bangor.ac.uk (H.-P.K.)

**Keywords:** VO_2_max, anti-inflammatory, machine learning, PCA

## Abstract

Relationships between demographic, anthropometric, inflammatory, lipid and glucose tolerance markers in connection with the fat but fit paradigm were investigated by supervised and unsupervised learning. Data from 81 apparently healthy participants (87% females) were used to generate four classes of fatness and fitness. Principal Component Analysis (PCA) revealed that the principal component was preponderantly composed of glucose tolerance parameters. IL-10 and high-density lipoprotein, low-density lipoprotein (LDL), and total cholesterol, along with body mass index (BMI), were the most important features according to Random Forest based recursive feature elimination. Decision Tree classification showed that these play a key role into assigning each individual in one of the four classes, with 70% accuracy, and acceptable classification agreement, κ = 0.54. However, the best classifier with 88% accuracy and κ = 0.79 was the Naïve Bayes. LDL and BMI partially mediated the relationship between fitness and fatness. Although unsupervised learning showed that the glucose tolerance cluster explains the highest quote of the variance, supervised learning revealed that the importance of IL-10, cholesterol levels and BMI was greater than the glucose tolerance PCA cluster. These results suggest that fitness and fatness may be interconnected by anti-inflammatory responses and cholesterol levels. Randomized controlled trials are needed to confirm these preliminary outcomes.

## 1. Introduction

In the 1950s, first observational evidence emerged showing that physically active individuals had a lower risk of cardiovascular disease (CVD) [1]. This evidence was later corroborated by the protective effect found for cardiorespiratory fitness (CRF), as shown in the Aerobics Center Longitudinal Study in 1989 [2,3]. Since then, several reviews, systematic reviews, and meta-analysis have confirmed and highlighted the protective role of CRF regardless the level of fatness [4,5,6,7]. According to the “fat but fit paradox”, people who have a high level of CRF may be better protected from the risk of CVD than leaner people who have low CRF [8]. However, only a small proportion of US citizens can be considered “fat and fit”, and obesity is independently associated with low CRF, simply because obese people are generally less active [9].

Lahoz-Garcia et al. [10] showed an interesting partial mediation of CRF between diet and obesity in schoolchildren, meaning that higher CRF contributes, for the same diet, to a lower fat mass (FM). Consistently, others have found that moderate to vigorous physical activity levels, thus higher CRF, were independently associated with a lower atherogenic index of plasma, namely blood fat strongly related with CVD, regardless of diet; and that central adiposity mediated, in other words explains, the relationship between moderate to vigorous physical activity levels and atherogenic index of plasma [11]. This would rule in favor of the protective role of higher CRF against CVD risk. Moreover, poor CRF has been associated with glucose intolerance [12] and a higher risk of insulin resistance in apparently healthy individuals [13]. Furthermore, it has been hypothesized that low CRF could provide an early sign of insulin resistance [14].

Obesity has been shown to be associated with low level systemic inflammation in connection with increased adipose tissue mass [15,16]. In turn there is evidence, in animal studies, of the possible role of inflammation on over-nutrition [17]. However, physical activity may counteract over-nutrition behavior at the hypothalamic level by means of anti-inflammatory signaling mediated interleukin-10 (IL-10) [17]. An anti-inflammatory role of IL-10 has been found also in rat skeletal muscle tissue [18]. In humans it was found consistently that intensive cycling is able to increase, 1 hour after the exercise, gene expression of several interleukins including IL-10, but not IL-6 [19]. High intensity exercise showed an acute, 30 minutes, IL-10 and IL-6 increase in overweight-obese inactive individuals, but this increase was not elicited by moderate intensity exercise [20]. Nevertheless, two weeks of high intensity exercise in overweight-obese unfit individuals did not show a chronic increase in IL-10 nor in IL-6 [21,22]. Rather, a chronic elevation of IL-10 found in obese women was reduced by 12 weeks of lifestyle intervention, including 30 minutes of exercise a day, only in those obese women who did not have metabolic syndrome [23]. Furthermore, higher serum concentration of IL-10 was found in older adults with a higher volume of physical activity [24]. Additionally, animal models show a possible protective role of anti-inflammatory signaling on cardiac function (i.e., left ventricular end-diastolic pressure) [25], a finding supported in human studies involving coronary heart disease patients, obese and diabetic individuals [26,27].

To further investigate the relationship between cardiovascular fitness and body composition characteristics i.e., fatness, we used a database, which combined demographic, blood lipids, insulin resistance, and inflammatory variables in association with CRF and FM% values. Our approach was to create a categorical variable composed of four classes, based on CRF and FM% levels. The four classes or categories are termed High Fatness with High Fitness (HFHF), High Fatness with Low Fitness (HFLF), Low Fatness with High Fitness (LFHF) and finally Low Fatness with Low Fitness (LFLF). The cutoff levels between categories were identified according to the literature [28,29]. We have applied a data driven approach consisting of four steps. First is an unsupervised learning phase, where the variables are clustered using Principal Component Analysis (PCA) [30]. PCA allows clustering of the variables into principal components. Second, a supervised learning phase was deployed to use those clusters in the feature importance selection. We opted for feeding the PCA components as well as the other variables into the feature importance selection algorithm because, although PCA combines uncorrelated variables with one another in such a way that each principal component will maximize variance, this does not mean that the components per se will be the most important classification features. Therefore, as a second step, we have used the same categorical four classes’ dependent variable for a random forest based feature importance selection. In detail, we have used the Boruta algorithm, which is an improvement of the Random Forest feature selection model, also known as recursive feature elimination [31,32]. The Boruta algorithm adds randomness to the importance evaluation algorithm, so that the certainty about the importance of a given variable is increased. In short, a randomized copy of the variables is made at each iteration of the random forest importance computation. Thus, if a variable has a higher importance than the maximal importance of all randomized attributes it is retained. If there is some uncertainty, or if a variable has a lower importance it is rejected or discarded [32].

Third, a decision tree was used in order to define the discriminating path to the four classes of fitness and fatness. This classification model was used to visualize which independent variables would best split the data points into the four classes. However, classification was not limited to the decision tree. Another four classification models were used as well with the intent of testing which classification model would maximize the use of the selected independent variables, or features. The four alternative machine learning classification models were Multiple Logistic Regression, Decision Tree, Naïve Bayes, and K-nearest neighbors. This step was necessary to test whether the features selected would effectively classify the data points. Finally, a fourth step, a mediation and moderation analysis [33] was conducted in order to investigate whether attenuation between CRF and FM% would occur when one of the variables extracted was used as covariate. We hypothesized that we would find attenuations, as previously shown in the literature [10,11], by means of variables linked to fat metabolism. The overall aim of this study was to use a data driven approach, employing machine-learning techniques, to generate new insights connecting fitness and fatness with demographic, blood lipids, insulin resistance, and inflammatory variables.

## 2. Materials and Methods

### 2.1. Study Design and Participants

The data analyzed in this study originated from two separate data collections conducted at Bangor University. Data from 81 apparently healthy participants (10 males and 71 females) were included in the analysis. All participants were informed about the study protocols and objectives, and provided written consent prior to the start of the studies. Study protocols were approved by the Ethics Committee of the School of Sports, Exercise and Health Sciences Department of Bangor University in conformity with the Declaration of Helsinki. The design of this study was purely observational.

### 2.2. Body Composition, Fat Mass Percentage, Blood Markers and Cardiorespiratory Fitness Assessment

Participants were pre-screened for cardiovascular diseases by means of the American Heart Association/American College of Sports Medicine Pre-Participation Questionnaire [34]. However, participants with elevated fasting levels of glucose, insulin and lipids were not per se excluded from this study. Body composition, fasting blood lipid profile and CRF (VO_2_max) were determined using standardized protocols described previously [21]. A cardiorespiratory fitness test was executed on a cycle ergometer (Corival 400, Lode, Groningen, The Netherlands), the protocol consisted of an incremental exercise test to exhaustion (1min at 50 + 20 W increments per minute). Oxygen uptake was measured breath by breath by means of a metabolic card (ZAN 600 CPET, Oberthulba, Germany). Fasting blood lipid profile (total Cholesterol, LDL and HDL), plasma insulin, plasma glucose, leptin and cytokines (IL-6, IL-10, and TNF-α) collection and analysis is also described in Sartor et al. [21]. Plasma glucose was analyzed by immobilized enzymatic assay (YSI 2300 STAT, Incorporated Life Sciences, Yellow Springs, OH, USA). Lipid profile was analyzed from plasma samples by optic enzymatic assay (Reflotron^®^, Roche Diagnostics, Mannheim, Germany). Plasma insulin was analyzed by ELISA (ultrasensitive human insulin ELISA kit, Mercodia, Uppsala, Sweden). Cytokines (IL-10, IL-6 and TNF-α) and adipokines were also analyzed from fasting plasma samples by ELISA (Bender MedSystems GmbH, Austria and BioVendor, Laboratoní medicína, Czech Republic, respectively). Insulin sensitivity and β-cell function were estimated using fasting plasma insulin and glucose by means of the Homeostatic model assessment 2 (HOMA2) [35].

### 2.3. Classification Criteria

Four classes were extracted from the database described above; a Higher-Fatness with Higher-Fitness (HFHF) group, a Higher-Fatness with Lower-Fitness (HFLF) group, a Lower-Fatness with Higher-Fitness (LFHF) group, and finally a Lower-Fatness with Lower-Fitness (LFLF) group. The grouping criteria were taken from Gallagher et al. [28] for fatness, and the American College of Sports Medicine guidelines [29] for fitness. The criteria are represented in Table 1.

### 2.4. Data Analytics

#### 2.4.1. Preprocessing

The full dataset collected at Bangor University premises was loaded into RStudio (Version 1.2.5033, 2009–2019 RStudio Inc., Boston, MA, USA). This initial dataset included 25 independent variables. A first missing data filter was applied and all variables with more than 70% missing data were discarded. After this step, 19 independent variables were retained. Two variables were converted into factorial variables, the classification variable as explained in Table 1 and the variable Sex. The retained variables were visualized to reveal imbalance. This visualization showed an imbalance towards females, as they represented 87% of our dataset. The imbalance was a consequence of the original research question of one data collection being confined to females. A zero- and near zero-variance predictors analysis was conducted, by means of nearZeroVar function (caret R package), to eliminate any independent variables that would not add anything in explaining variance (Table 2). However, no variables were rejected based on these criteria [36]. The preProcess function (caret R package) was used to center and scale the variables and missing data, were imputed using the bagImpute function which uses the bootstrap aggregating method [37]. Outliers were detected as values outside boxplot notches, using boxplot function (graphics R package). The notches were set as the median, plus or minus the standard error [38]. The detected outliers were excluded from the analysis.

#### 2.4.2. Principal Component Analysis and Feature Selection

Once the data were pre-processed a principal component analysis was conducted to find what combination of variables would explain the variability of the data. The function PCA (FactoMineR R package) as described in [39] was used. Eigenvalues, which represent the amount of the variation explained by each principal component, were extracted by fviz_eig. The number of retained components was set so that 70% of the total variance is explained. Correlation plots of all variables were produced using the corrplot function (corrplot R package). The importance of the twenty variables including five new Principal Components was evaluated by a recursive feature elimination technique based on the Boruta Random Forest method (Boruta R package) [32]. The Boruta function compares original importance attributes against importance achievable by shadow random variables, in iterations until convergence. The principal components were also included in the feature selection step, to test whether the most variation corresponded with the highest importance. 

#### 2.4.3. Decision Tree

A decision tree was built using the nine variables selected by the Boruta algorithm, with the exclusion of the PCA dimensions. As first step, the class imbalance was compensated by means of weights for simple random sample (i.e., 1/probability). The decision tree was constructed using the rpart function (rpart R package) and vitalized by rpart.plot (rpart.plot R package). Tree depth was set as the smallest tree within one standard error of the minimum cross validation error [40].

#### 2.4.4. Classification Models

Multiple logistic regression, decision tree, naïve Bayes, and κ-nearest neighbors classification model were trained on our dataset by means of the train function (caret R package) as described in Kuhn [36]. The classes were the four subgroups (HFHF, HFLF, LFHF, LFLF) described above. In order to perform the multinomial logistic regression, the multinom method was selected within the train function. In order to evaluate the performance of each single classifier, accuracy tables and confusion matrices were generated, using the confusionMatrix in caret and visualized thanks to ggplot (ggplot2 R package) [36].

#### 2.4.5. Mediation and Moderation Analysis

Mediation analysis was conducted by means of the mediation R package [41]. Before analyzing, the mediation and moderation raw data for each variable were assessed for normality and linearity by means of quantile-quantile plots (qqnorm function, from the basic stats R package), centered, and scaled when required, as described earlier. Linear regressions models, via the lm function (stats R package), were built between the mediator and the independent variable (relative VO_2_max), and between the dependent variable (Fat Mass percentage) and the independent variable-mediator combined. The mediate function simulated the comparison between these two linear regressions, showing if the mediation would add a significant contribution in relating the independent and dependent variables. The mediation analysis resulted in the Average Causal Mediation Effects (ACME), the Average Direct Effects (ADE), and the combined effects (Total Effect), and the proportion mediated (Prop. Mediated). Moderation was executed by the gylma and stargazer R packages. A linear model was built between the dependent variable and independent variable plus the moderator, and between the dependent variable and the moderator plus the product.

### 2.5. Statistical Analysis

The descriptive statistics, means and standard deviations of all participants for the 15 included variables and for each of the four subgroups were analysed using the arsenal R package [42]. Data for the four subgroups were split using the filter function supported by the dplyr R package. One-way ANOVAs were performed to compare the four sub-groups and they were followed-up when appropriate both by the tableby function (arsenal R package). Significance level was set at 0.05.

## 3. Results

### 3.1. Subgrouping and Difference Analysis

As described in the method section, four subgroups were derived according to participants’ CRF, body FM%, age, and sex. The subgroups sizes are not evenly distributed. Two subgroups HFHF and LFHF are rather small (*N* = 9, *N* = 6, respectively). In line with our intention to form four groups of different fatness and fitness levels, the ANOVA and follow-up showed significant differences between the two higher-fitness and lower-fitness levels. Moreover, the HFHF group and the LFHF groups also showed a significant difference in fitness, the lower in fatness being fitter (40.1 ± 2.9 mL/kg/min) than the higher in fatness (34.3 ± 4.3 mL/kg/min). As for the higher fatness/lower fatness split, this was fully achieved, as confirmed by the ANOVA and follow-ups (Table 3). As to be expected, BMI was significantly higher in the HFLF group compared with the LFHF and LFLF subgroups. There was a trend towards a higher BMI for the HFLF group when compared with the HFHF group, and a trend towards a higher BMI in the HFHF group compared with the LFHF group. It is to be noted that BMI does not fully reflect FM% (Table 3). Total fasting plasma Cholesterol levels showed significantly higher levels in the HFLF compared with the HFHF and LFHF groups. There was a strong trend towards a higher cholesterol level in the LFLF group compared with the HFHF group. The LFHF group showed higher HDL than the HFHF group. The LFLF group had a higher HDL level than the HFLF group. Moreover, there were two strong trends for a higher HDL in the LFLF group and the LFHF group versus the HFHF and the HFLF groups, respectively. LDL was higher in the HFLF group compared with the HFHF, LFHF, and LFLF groups. Finally, fasting plasma insulin was higher in the HFLF compared with the HFHF. Interestingly two, LFHF and LFLF, groups showed higher insulin values than the HFHF group (Table 3).

### 3.2. Principal Component Analysis

The independent variables, once filtered for missing data, were clustered by means of principal component analysis. Five principal component dimensions were found that explained 70% of the variance (Figure 1). Dimension 1 was dominated by glucose tolerance features, dimension 2 by Leptin and Sex, dimension 3 was constituted by lipid profile, dimension 4 by triglycerides and glucose, and, finally, dimension 5 by BMI and weight. (Figure 1). In Figure 2 the classification and the weight of the single individuals is shown when the first two components are put in relation.

These five dimensions were further included in the feature selection process. Recursive feature elimination based on random forest showed that the stronger features in describing the four groups were IL-10, BMI, total cholesterol, HDL, LDL, dimension 1, beta cell function, dimension 4, IL-6, Age, dimension 3, and weight. In Figure 2 the interrelationship of the first two PCA components is shown and the four groups are clustered. Fitter groups tend to develop along dimension 1 while the less fit along dimension 2.

### 3.3. Classification Models

The Random Forest based recursive feature elimination Boruta algorithm found twelve variables as certainly important in classifying the four fatness and fitness classes (Figure 3). Amongst these twelve are PCA dimensions 1,4 and 3, in order of importance. While the algorithm is uncertain about dimension 5 and discards dimension 2. IL-10, BMI, and cholesterol levels are clearly the most important variables. In Appendix A the first 10 selected variables are shown as boxplot. Additionally, in Appendix A linear correlations between variables are displayed, showing how the retained variables still carry most of the correlations.

When the twelve variables, including the PCA dimensions, selected by the Boruta importance algorithm were used to generate the classification model, we found acceptable classification performances. In fact, the Multiple Logistic Regression model showed a classification accuracy of 0.77 (95% CI: 0.6717, 0.8627), significantly higher than the No Information Rate (0.4691), and a κ-coefficient of 0.65, Figure 4. The Decision Tree model, displayed in Figure 5, although having the lowest accuracy (0.70, 95% CI: 0.5919, 0.8001) amongst the models generated here, still had an accuracy significantly higher than its No Information Rate (0.432), and an acceptable κ-coefficient (0.54) (Figure 4). The Naïve Bayes classifier showed the highest accuracy (0.88, 95% CI: 0.7847, 0.9392), significantly higher than the No Information Rate (0.58), and a moderate κ-coefficient equal to 0.79 (Figure 4). Finally, the K-Nearest Neighbors classifier had an accuracy of 0.73 (95% CI: 0.6181, 0.8213), which was, however, not higher than the No Information Rate (0.76), with a rather weak agreement, a κ-coefficient of 0.47 (Figure 4). Overall, the latter performed worse than the other classification models.

### 3.4. Mediation and Moderation Analysis

All selected variables were analyzed for mediation and moderation. As shown by the quantile-quantile plots in Figure 6, LDL and BMI did not require further scaling and/or centering and were the only two variables to show a significant partial mediation effect between CRF and FM% (Figure 7). Details of the causal mediation analysis are captured in Table 4.

## 4. Discussion

This present study embraces artificial intelligence as a tool to provide new insight into the fat but fit paradox [8]. Using unsupervised and supervised machine learning approaches to interrogate existing physiological data, this work indicates connection between markers of dyslipidemia, inflammation and cardiorespiratory fitness that reveal possible functional interaction of physiological systems underpinning the “fat but fit paradox”.

### 4.1. Descriptive Statistics in Relation to Fatness and Fitness

We have created four classes, or groups, in line with population normative cut-off values [28,29]. Consistently, these groups differed significantly from one another in terms of fitness and fatness (Table 3). Fasting total cholesterol levels and LDL were significantly higher in the HFLF group, while HDL was higher in the groups with lower fatness. The decision tree depicted in Figure 5 shows how well HDL and LDL alone could differentiate the HFHF group from the other groups. Although IL-10 did not show significant differences between the four groups, whereas IL-6 did, IL-10 seemed to be involved in the differentiation of individuals with lower fitness level in function of their fatness (Figure 5). Moreover, the Analysis of Variance amongst the four groups also showed differences in fasting insulin levels. Fasting insulin was the highest in the HFLF group and the lowest in the HFHF group (Table 3). This is of particular interest because it seemed to be associated with fitness rather than with fatness levels. Fitness has been shown to play an important role in protecting against glucose intolerance [43]. This may be related to the well-known effect of muscle contractile activity, hence exercise training, on insulin sensitivity [44].

### 4.2. Machine Learning

Principal component analysis clustered the various markers available in this study so that they could better explain the variance of the fatness and fitness categorical variable. This resulted in PCA Dimension 1, mainly composed of glucose tolerance indicators, such as fasting insulin, insulin sensitivity, and insulin resistance, as well as beta cell function derived from the HOMA2 model (Figure 1). However, supervised learning, namely the random Forest based feature selection algorithm, revealed that the importance of IL-10, cholesterol levels (i.e., HDL, LDL and total Cholesterol) along with BMI in classifying the four classes was greater than that of the above mentioned glucose tolerance PCA cluster. The interesting aspect of our approach is that our analysis clearly points towards dominant features, namely IL-10, LDL, HDL, BMI, for categorizing our four groups, in competition with other features, which are just as well known to be influenced by fatness and fitness. Besides the potential exercise dependent link between IL-10 and insulin/leptin sensitivity in the hypothalamus in animal studies [17], exercise was found to increase IL-10 levels in overweight-obese human subjects [20]. An interlink between fatness and IL-10, however, was found in obese subject after weight loss, revealing higher IL-10 levels [45]. Therefore, distinct features of our data could point towards an important discriminating function of IL-10 and LDL/BMI for fatness and fitness classification and could be linked to these findings. Moreover, exercise has been found to effect LDL as well as HDL levels [46].

### 4.3. Partial Mediation

Fatness and fitness are significantly inversely related [47]. This was confirmed by our data. In addition to this, however, we found that CRF is indirectly related to FM% through the mediation of LDL and BMI. Previous literature found that BMI could mediate CRF and cardio-metabolic risk in schoolchildren [48]. Another investigation in schoolchildren using a large dataset showed that CRF may have a beneficial effect on lipid profile, insulin metabolism and inflammation independent of fatness [49]. Our results seem to lead in the same direction. Specific effects of exercise training, of a high enough intensity, to promote aerobic capacity improvements have been linked to a decrease in concentration of atherogenic ox-LDL [46,50]. In addition, upregulation of fatty acid metabolism and transport through exercise dependent signaling pathways (particularly peroxisome proliferator-activated receptor) [51,52] and concurrent alterations in lipid profiles [53,54] are well described. Interestingly, IL-10 was found to be linked with LDL level as IL-10 was shown to induce uptake of LDL by fluid-phase endocytoses in macrophages leading to lowered LDL plasma levels [55].

### 4.4. Implications

We are aware that our study is retrospective. Thus, it provides a limited level of evidence. It is beyond the purpose of this study to accept or not the hypothesis that fitness plays a protective role in people with higher level of fatness. Yet the discriminating role that anti-inflammatory and cholesterol levels seem to make sense when addressing fatness and fitness may point in the direction of “healthy obesity” when the CRF level is high [4]. 

### 4.5. Limitations

The current study is based on data from 81 individuals. Several variables, such as systolic and diastolic Blood Pressure and C-Reactive Protein, had to be excluded from the analysis because of missing data. The observations and conclusions drawn from this study would need to be verified in a larger dataset. This study does not provide direct experimental evidence, but is merely observational and retrospective. These considerations need to be taken into account when evaluating our results and conclusions. Our dataset has more females than males, and although sex did not appear to play a key role in determining the classification, we cannot exclude that, with a higher number of males this factorial variable would or could have had a greater weight. Finally, by dividing our dataset into four classes we observed that these were not evenly distributed. This issue was partially mitigated by balancing, using class weights.

## 5. Conclusions

Our data analytics approach has shown a potential key role of IL-10 as well as HDL, LDL, total Cholesterol and BMI in the classification of people according to their fatness and fitness levels. Unsupervised learning showed that a cluster of glucose tolerance related variables explains the highest quote of the variance of the categorical variable. However, supervised learning did not select this PCA cluster. Mediation analysis showed that LDL and BMI partially explain the association between fitness and fatness. These results suggest that CRF and FM% may be interconnected by anti-inflammatory responses and cholesterol blood levels. This may be in line with the protective role of cardiorespiratory fitness suggested in recent years. However, large randomized controlled trials are needed to validate this hypothesis experimentally and conclusively.

## Figures and Tables

**Figure 1 ijerph-18-01800-f001:**
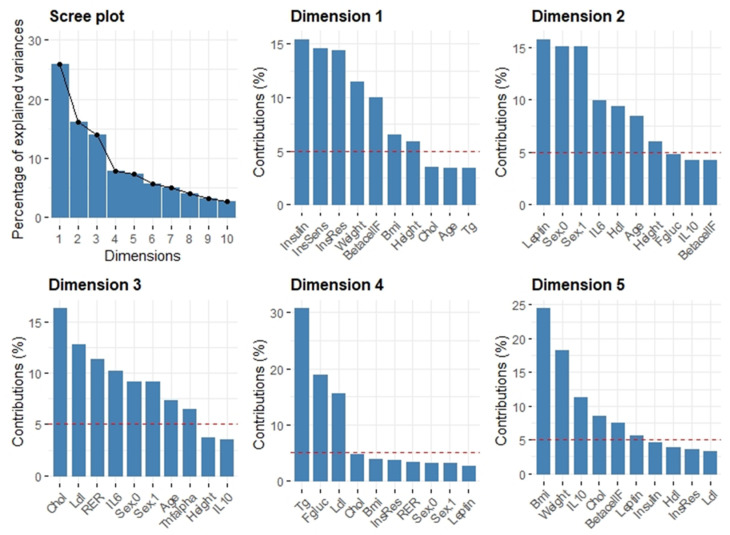
Output of the principal component analysis: BMI = Body Mass Index, Chol = Fasting Total Cholesterol, HDL = Fasting High Density Lipoprotein, LDL = Fasting Low Density Lipoprotein, TG = Fasting TriGlycerides, Fgluc = Fasting Glucose, BetacellF = β cell Function, InsSens = Insulin Sensitivity, InsRes = Insulin Resistance, TNFalpha = Tumor Necrosis Factor α, IL-6 = Interleukin-6, IL-10 = Interleukin-6, RER = Respiratory Exchange Ratio, Sex.0 = females, Sex.1 = males.

**Figure 2 ijerph-18-01800-f002:**
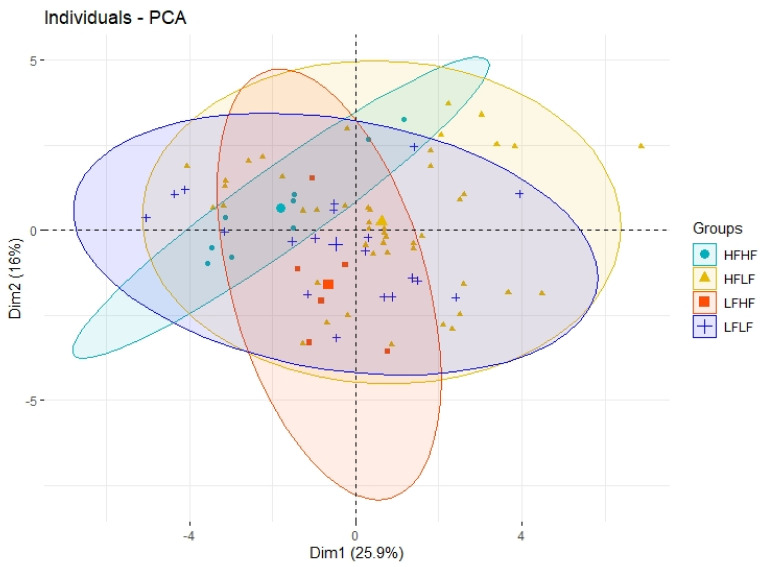
Clustering of the categorical variable, including the four fatness and fitness permutations. Relationship between the first principal component and the second principal component computed by PCA. The size of the icons for the single individuals shows their weight in classification. HFHF = Higher-Fatness with Higher-Fitness group, HFLF = Higher-Fatness with Lower-Fitness group, LFHF = Lower-Fatness with Higher-Fitness group, LFLF = Lower-Fatness with Lower -Fitness group.

**Figure 3 ijerph-18-01800-f003:**
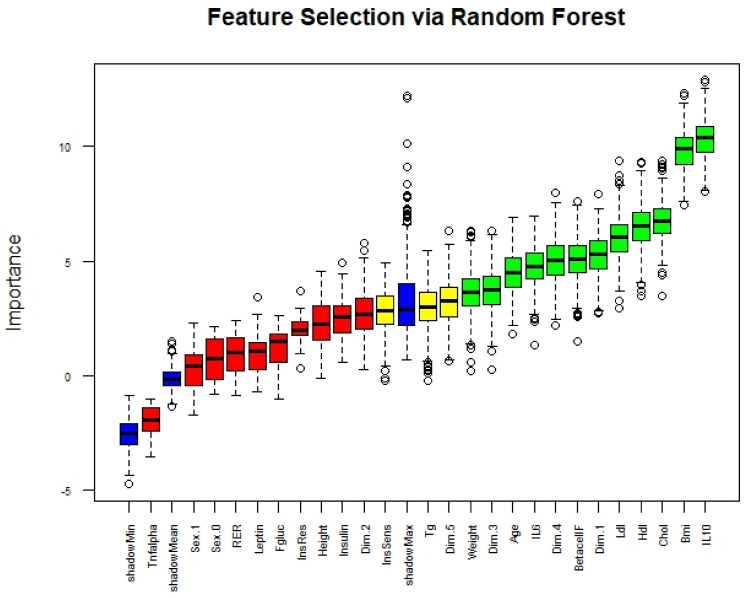
Recursive feature elimination; in green are depicted the variables that are certain. IL6. = Interleukin-6, IL-10 = Interleukin-6, RER = Respiratory Exchange Ratio, Sex.0 = females, Sex.1 = males, Dim.1 = Dimension 1 of the PCA, Dim.2 = Dimension 2 of the PCA Dim.3 = Dimension 3 of the PCA, Dim.4 = Dimension 4 of the PCA, Dim.5 = Dimension 5 of the PCA.

**Figure 4 ijerph-18-01800-f004:**
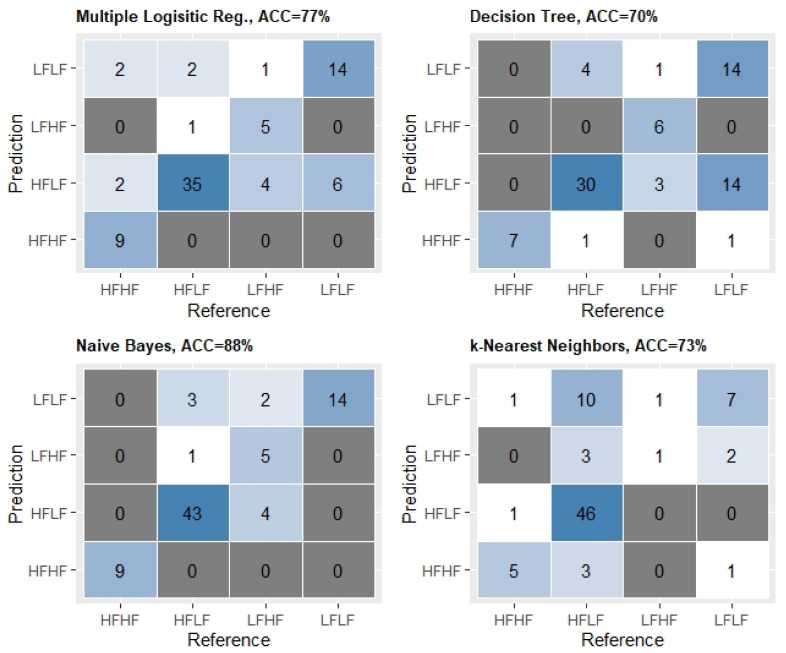
Confusion Matrices, and accuracy of the four classification models. ACC = accuracy, HFHF = Higher-Fatness with Higher-Fitness group, HFLF = Higher-Fatness with Lower-Fitness group, LFHF = Lower-Fatness with Higher-Fitness group, LFLF = Lower-Fatness with Lower -Fitness group.

**Figure 5 ijerph-18-01800-f005:**
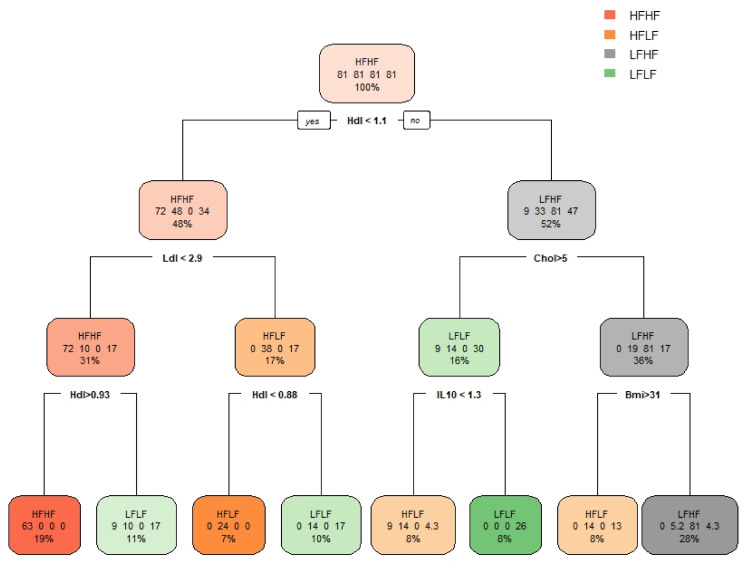
Decision tree, where HDL = High Density Lipoprotein, LDL = Low Density Lipoprotein, IL-10 = Interleukin-10 and BMI = Body Mass Index are expressed in their original dimensions (mmol/L, mmol/L, pg/mL, respectively). HFHF = Higher-Fatness with Higher-Fitness group, HFLF = Higher-Fatness with Lower-Fitness group, LFHF = Lower-Fatness with Higher-Fitness group, LFLF = Lower-Fatness with Lower -Fitness group.

**Figure 6 ijerph-18-01800-f006:**
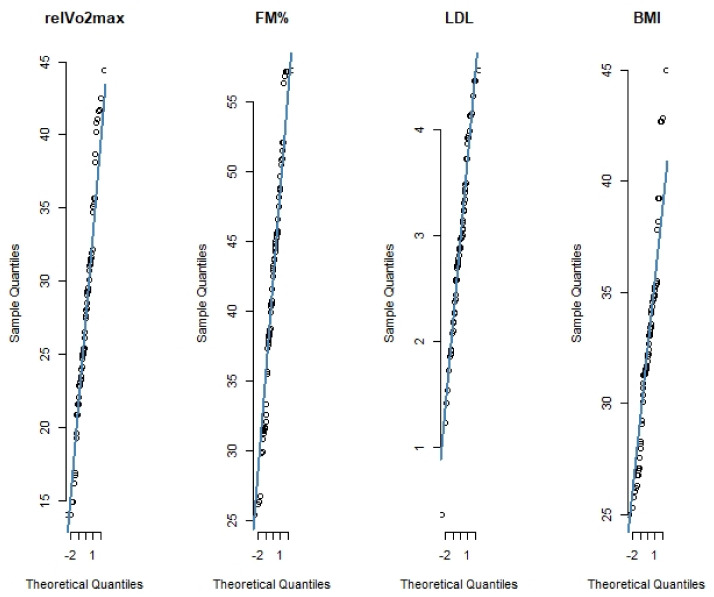
Quantile–quantile plots of the variables that showed partial mediation.

**Figure 7 ijerph-18-01800-f007:**
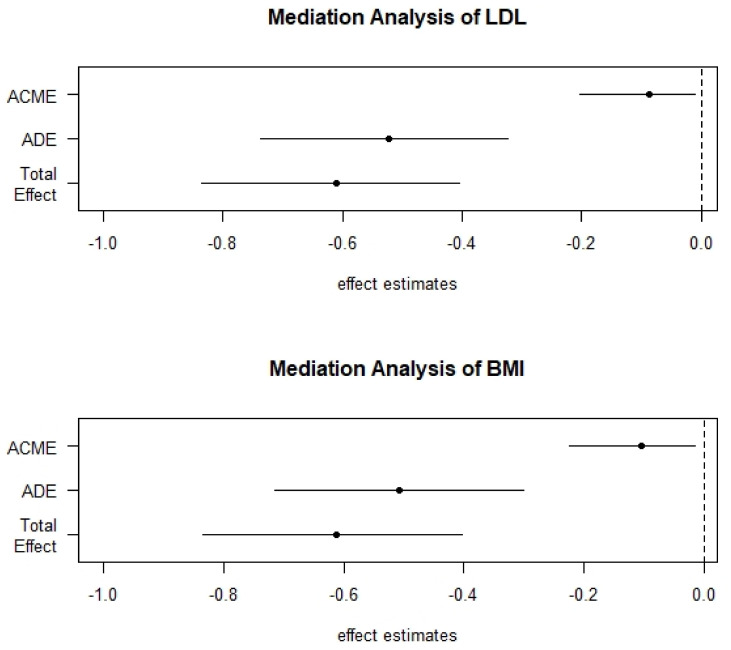
Decomposed Mediation Analysis plot: ACME = Average Causal Mediation Effect, ADE = Average Direct Effect, LDL = Low Density Lipoprotein, BMI = Body Mass Index.

**Table 1 ijerph-18-01800-t001:** Classification criteria for body fat percentage and relative VO_2_max (mL/kg/min), age and sex.

Age	Males	Females
Young	if AGE < 40 years AND if Sex = 1 AND FatMass% ≥ 26 then Higher-Fatness	Elseif Sex = 0 AND FatMass% ≥ 39 then Higher-Fatness
Middle-Age	if 59 ≥ AGE ≥ 40 AND if Sex = 1 AND FatMass% ≥ 29 then Higher-Fatness	Elseif Sex = 0 AND FatMass% ≥ 41 then Higher-Fatness
Older	if AGE ≥ 60 AND if Sex = 1 AND FatMass% ≥ 31 then Higher-Fatness	Elseif Sex = 0 AND FatMass% ≥ 43 then Higher-Fatness
Young/Middle/Older	Else Lower-Fatness	Else Lower-Fatness
Young	If AGE < 29 AND if Sex = 1 AND if relVO_2_max > 45.7 then Higher-Fitness	Elseif Sex = 0 AND if relVO_2_max > 39.5 then Higher-Fitness
Middle-Age	If 39 ≥ AGE > = 30 AND if Sex = 1 AND if relVO_2_max > 44.4 then Higher-Fitness	Elseif Sex = 0 AND if relVO_2_max > 36.7 then Higher-Fitness
	If 49 ≥ AGE ≥ 40 AND if Sex = 1 AND if relVO_2_max > 42.4 then Higher-Fitness	Elseif Sex = 0 AND if OrelVO_2_max > 35.1 then Higher-Fitness
Older	If AGE > 50 AND if Sex = 1 AND if relVO_2_max > 38.3 then Higher-Fitness	Elseif Sex = 0 AND if OrelVO_2_max > 31.4 then Higher-Fitness
Young/Middle/Older	Else Lower-Fitness	Else Lower-Fitness

**Table 2 ijerph-18-01800-t002:** Zero- and near zero-variance predictors analysis.

	Frequency Ratio	Percent Unique	Zero Variance	Near Zero Variance
Sex	7.100000	2.469136	FALSE	FALSE
Age	1.555556	23.456790	FALSE	FALSE
Height	1.142857	28.395062	FALSE	FALSE
Weight	1.000000	77.777778	FALSE	FALSE
BMI	1.500000	75.308642	FALSE	FALSE
Chol	1.000000	72.839506	FALSE	FALSE
HDL	1.000000	62.962963	FALSE	FALSE
LDL	1.333333	69.135802	FALSE	FALSE
TG	5.250000	49.382716	FALSE	FALSE
Fgluc	1.500000	62.962963	FALSE	FALSE
Leptin	1.000000	76.543210	FALSE	FALSE
Insulin	1.000000	77.777778	FALSE	FALSE
BetacellF	1.000000	77.777778	FALSE	FALSE
InsSens	1.000000	80.246914	FALSE	FALSE
InsRes	1.000000	43.209877	FALSE	FALSE
TNFalpha	1.333333	66.666667	FALSE	FALSE
IL-6	1.333333	71.604938	FALSE	FALSE
IL-10	1.000000	50.617284	FALSE	FALSE
RER	1.200000	34.567901	FALSE	FALSE

BMI = Body Mass Index, Chol = Fasting Total Cholesterol, HDL = Fasting High Density Lipoprotein, LDL = Fasting Low Density Lipoprotein, TG = Fasting TriGlycerides, Fgluc = Fasting Glucose, BetacellF = β cell Function, InsSens = Insulin Sensitivity, InsRes = Insulin Resistance, TNFalpha = Tumor Necrosis Factor α, IL-6 = Interleukin-6, IL-10 = Interleukin-6, RER = Respiratory Exchange Ratio.

**Table 3 ijerph-18-01800-t003:** Descriptive Statistics of Database, difference analysis and follow-up analyses.

	HFHF (*N* = 9)	HFLF (*N* = 47)	LFHF (*N* = 6)	LFLF (*N* = 19)	Total (*N* = 81)	ANOVA *p* Value	*t*-Test Follow-Up HFHF vs. HFLF *p* Value	*t*-Test Follow-Up HFHF vs. LFHF *p* Value	*t*-Test Follow-Up HFHF vs. LFLF *p* Value	*t*-Test Follow-Up HFLF vs. LFHF *p* Value	*t*-Test Follow-up HFLF vs. LFLF *p* Value	*t*-Test Follow-Up LFHF vs. LFLF *p* Value
Relative VO_2_max (mL/kg/min)						**<0.001**	**<0.001**	**0.013**	**<0.001**	**<0.001**	0.346	**<0.001**
Mean(SD)	34.349 (4.237)	25.492 (6.432)	40.123 (2.992)	26.968 (3.272)	27.906 (6.932)							
Range	29.560–42.530	14.050–41.700	35.680–44.400	19.300–31.500	14.050–44.400							
Fat Mass %						**<0.001**	0.108	**0.003**	**0.003**	**<0.001**	**<0.001**	0.065
Mean(SD)	41.951 (5.686)	45.842 (6.686)	31.395 (4.828)	35.548 (4.510)	41.925 (7.871)							
Range	32.100–47.500	29.800–57.240	25.400–37.750	26.180–40.500	25.400–57.240							
Age, yrs						0.063						
Mean (SD)	42.444 (7.764)	34.787 (13.454)	24.500 (8.666)	33.526 (12.624)	34.580 (12.866)							
Range	33.000–50.000	19.000–57.000	19.000–42.000	20.000–49.000	19.000–57.000							
BMI						**0.003**	0.063	0.099	0.687	**0.009**	**0.005**	0.454
Mean (SD)	31.174 (1.572)	33.728 (3.949)	29.165 (2.828)	30.577 (4.217)	32.367 (4.061)							
Range	27.580–33.080	26.970–44.990	25.000–31.440	25.300–39.230	25.000–44.990							
Height, m						0.894						
Mean (SD)	1.671 (0.114)	1.662 (0.092)	1.657 (0.047)	1.681 (0.099)	1.667 (0.093)							
Range	1.570–1.950	1.500–1.950	1.580–1.710	1.540–1.950	1.500–1.950							
Weight, kg						0.138						
Mean (SD)	87.458 (13.453)	93.392 (14.219)	80.335 (10.806)	87.141 (18.855)	90.299 (15.427)							
Range	67.990–119.050	63.400–125.690	62.500–91.630	61.750–125.690	61.750–125.690							
Cholesterol, mmol/L						**0.006**	**0.003**	0.995	0.055	**0.013**	0.447	0.113
Mean (SD)	3.839 (0.623)	4.756 (0.830)	3.837 (0.753)	4.573 (1.003)	4.543 (0.904)							
Range	3.210–5.020	2.590–6.260	2.830–4.970	3.170–6.320	2.590–6.320							
HDL, mmol/L						**0.008**	0.900	**0.025**	0.056	0.057	**0.004**	0.873
Mean (SD)	1.023 (0.205)	1.040 (0.395)	1.368 (0.327)	1.407 (0.554)	1.149 (0.445)							
Range	0.610–1.420	0.370–2.490	1.110–1.790	0.700–2.590	0.370–2.590							
LDL, mmol/L						**<0.001**	**0.002**	0.367	0.388	**<0.001**	**0.037**	0.186
N-Miss	0	1	0	0	1							
Mean (SD)	2.456 (0.509)	3.221 (0.667)	2.165 (0.699)	2.769 (1.006)	2.949 (0.816)							
Range	1.860–3.240	1.910–4.460	1.230–2.890	0.360–4.560	0.360–4.560							
Fglucose, mmol/L						0.237						
N-Miss	0	0	0	1	1							
Mean (SD)	5.416 (0.652)	5.078 (0.802)	4.665 (0.565)	5.030 (0.351)	5.074 (0.700)							
Range	4.710–6.360	3.850–9.050	3.800–5.300	4.190–5.460	3.800–9.050							
Leptin, ng /mL						0.118						
N-Miss	0	0	1	0	1							
Mean (SD)	16.966 (10.307)	29.711 (15.740)	22.058 (18.130)	29.883 (16.539)	27.840 (15.895)							
Range	1.380–26.760	2.690–59.970	3.070–48.840	5.470–57.640	1.380–59.970							
Insulin, pmol/L						**0.040**	**0.012**	**0.023**	**0.039**	0.630	0.184	0.722
N-Miss	0	0	1	0	1							
Mean (SD)	5.667 (2.978)	11.053 (6.038)	9.722 (2.363)	9.021 (4.129)	9.881 (5.415)							
Range	1.730–9.640	1.210–28.090	7.350–12.400	2.410–17.470	1.210–28.090							
TNFalpha, pg/mL						0.992						
Mean (SD)	1.411 (1.669)	1.476 (2.253)	1.188 (1.952)	1.432 (2.046)	1.437 (2.093)							
Range	0.280–4.920	0.240–10.900	0.270–5.170	0.240–7.070	0.240–10.900							
IL-6, pg/mL						**0.045**	0.087	0.418	0.183	**0.028**	0.394	0.061
Mean (SD)	1.609 (0.525)	1.118 (0.811)	1.933 (0.983)	1.294 (0.589)	1.274 (0.777)							
Range	0.800–2.200	0.000–3.120	0.380–3.040	0.190–2.250	0.000–3.120							
IL-10, pg/mL						0.138						
N-Miss	0	0	1	0	1							
Mean (SD)	0.864 (0.224)	0.841 (0.332)	1.130 (0.848)	1.108 (0.662)	0.925 (0.470)							
Range	0.430–1.190	0.030–1.700	0.030–2.370	0.040–2.250	0.030–2.370							

HFHF = Higher-Fatness with Higher-Fitness group, HFLF = Higher-Fatness with Lower-Fitness group, LFHF = Lower-Fatness with Higher-Fitness group, LFLF = Lower-Fatness with Lower-Fitness group, VO_2_max = maximal oxygen uptake, BMI = Body Mass Index, HDL = Fasting High Density Lipoprotein, LDL = Fasting Low Density Lipoprotein, TNFalpha = Tumor Necrosis Factor α, IL-6 = Interleukin-6, IL-10 = Interleukin-6. Significant p-levels are highlighted in bold.

**Table 4 ijerph-18-01800-t004:** Causal Mediation Analysis, Quasi-Bayesian Confidence Intervals.

	Estimate	95% CI Lower	95% CI Upper	*p*-Value
ACME (LDL)	−0.0843	−0.1813	−0.01	0.024 *
ADE (LDL)	−0.5221	−0.7414	−0.30	<0.001 ***
Total Effect LDL)	−0.6063	−0.8271	−0.40	<0.001 ***
Prop. Mediated (LDL)	0.1308	0.0164	0.31	0.024 *
ACME (BMI)	−0.1078	−0.2205	−0.02	0.012 *
ADE (BMI)	−0.4996	−0.7034	−0.30	<0.001 ***
Total Effect (BMI)	−0.6075	−0.8211	−0.40	<0.001 ***
Prop. Mediated (BMI)	0.1728	0.0397	0.36	0.012 *

LDL = Low Density Lipoprotein, BMI = Body Mass Index, ACME = Average Causal Mediation Effect, ADE = Average Direct Effect, Prop. Mediated = Proportion of the effect Mediated. Significant values: *** <0.001, * <0.05. *N* = 81, Simulations: 1000.

## Data Availability

The data presented in this study are available on request from the corresponding author. The data are not publicly available due to restrictions related to the data protection regulations.

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
