# Peer review of "Plasma Interleukin-10 and Cholesterol Levels May Inform about Interdependences between Fitness and Fatness in Healthy Individuals"

_ijerph, 2021, doi:10.3390/ijerph18041800_

Round 1

Reviewer 1 Report

The overall aim of this study was to use a data driven approach, employing machine-learning techniques, to generate new insights connecting fitness and fatness with demographic, blood lipids, insulin resistance, and inflammatory variables. 

Hypothesis:  We hypothesized that we would find attenuations by means of variables linked to fat metabolism.

Overall, there are a lot of data in this article and some of the main points are lost. 

  • Aims and hypothesis should be at the end of the intro.
  • Line 77 sentence beginning with “This because…” should not be rephrased, a little confusing
  • Line 113, briefly explain CRF protocol
  • Statistical analysis area should be better organized. Make sure to have Statistical analysis and then use subcategories.  An example  is there is logistic equations used (line 176) and an are for statistical analysis line 201.  Make sure that is located in one place.
  • Please briefly summarize the mechanisms used instead of just referencing them. This would help the reader understand the mechanisms used instead of looking up the reference.

Author Response

Point 1: Hypothesis:  We hypothesized that we would find attenuations by means of variables linked to fat metabolism. Overall, there are a lot of data in this article and some of the main points are lost. Aims and hypothesis should be at the end of the intro.

Response 1: Thank you for the point of have raised. We agree that it will help clarity having both aims and hypothesis at the end of the introduction. We have now addressed this point in the text. 

Point 2: Line 77 sentence beginning with “This because…” should not be rephrased, a little confusing

Response 2: We agree, that sentence was indeed not clear. We have now rephrased it.

Point 3: Line 113, briefly explain CRF protocol

Response 3: We have added a short explanation of the CRF, a more extensive explanation of the test can be found in Sartor et al., 2010.

Point 4: Statistical analysis area should be better organized. Make sure to have Statistical analysis and then use subcategories.  An example  is there is logistic equations used (line 176) and an are for statistical analysis line 201.  Make sure that is located in one place.

Response 4: Although we understand the reviewer's point we would like to justify and possibly hold on our choice to separate the pre-processing and modeling part from the statistical analysis. Although, we did use logistic regression, in this context it was used as one of the classification models and not as a pure statistical regression model. At the moment under the paragraph 2.5 Statistical analysis we have collected only inferential statistical tests, such as ANalysis Of VAriance. 

Point 5: Please briefly summarize the mechanisms used instead of just referencing them. This would help the reader understand the mechanisms used instead of looking up the reference.

Response 5: It is a little unclear to us what are the mechanisms the reviewer is referring to. Are these the data analytic tools?

Reviewer 2 Report

The manuscript titled “Plasma interleukin-10 and cholesterol levels may inform about interdependences between fitness and fatness in healthy individuals” provide important and interesting information regarding the link between IL-10 land cholesterol in healthy individuals that could be classified between the fitness and fatness area. The manuscript is well written and scientifically sound, and the provided methods were designed to prove the central hypothesis. I just have to add some minor comments to make this manuscript suitable for publication in IJERPH.  

INTRODUCTION

1. Lines 54-58: Although IL-10 is a well-known anti-inflammatory cytokine, there is evidence indicating that not all obesity-associated studies have shown increased levels of IL-10 in individuals subjected to physical activity. I respectfully suggest that the authors add some of this evidence to provide a broader paradigm of IL-10 limitations. For instance, a quick search brought to me these manuscripts (authors are encouraged to read them but it is not mandatory to include them): 

    • DOI: 10.1016/j.cyto.2018.05.035
    • DOI: 10.1016/j.cyto.2015.10.003
    • DOI: 10.1210/jc.2002-021437 

2. Lines 73-100: This section looks like a methodological section. Thus, authors could add all of this explanation to materials and methods or rewrite them as an initial application of the methods used in this research, but all of them should be before Lines 62-72, since most research articles contain the objective and hypothesis at the end of the introduction. 

MATERIALS AND METHODS

3. The authors have indicated that participants were apparently healthy. Did the authors conduct at least an overall analysis to confirm if they were healthy?

4. Lines 107-109: Please provide the Ethics Committee Approval ID. 

5. Lines 115-116: Please provide the full details from the ELISA test. This is important since detection limits vary highly between manufacturers. 

RESULTS AND DISCUSSION

6. Line 239: Table 4 is mentioned here, but it is located very far away from this section. Authors should rearrange some figures or tables, so this table could be easily located. For instance, Fig. 2 and Fig. 3 could be merged in 1 figure. This is entirely a decision from the authors. 

Author Response

Point 1: Lines 54-58: Although IL-10 is a well-known anti-inflammatory cytokine, there is evidence indicating that not all obesity-associated studies have shown increased levels of IL-10 in individuals subjected to physical activity. I respectfully suggest that the authors add some of this evidence to provide a broader paradigm of IL-10 limitations. For instance, a quick search brought to me these manuscripts (authors are encouraged to read them but it is not mandatory to include them): 

    • DOI: 10.1016/j.cyto.2018.05.035
    • DOI: 10.1016/j.cyto.2015.10.003
    • DOI: 10.1210/jc.2002-021437 

Response 1: We thank the reviewer for pointing this aspect out. We have looked into the three references provided. They are actually very pertinent. We noticed that Barry et al., 2018 is consistent with our previous research, not fining chronic increase of IL-10 or IL-6 after two weeks of HIIT in unfit obese individuals, see also Sartor et al., 2010. However, an acute elevation of IL-10 due to exercise  is to be expected has shown by the reference we provided (Nieman et al., 2006 ) and by the one the reviewer kindly provided (Dorneles et al., 2016). It is also interesting to report a possible independent effect of metabolic syndrome on IL-10 circulating levels as shown by Esposito et al., (2003), which seems to strengthen the hypothesis that IL-10 may play a key protective role in individuals with higher CRF levels. We have added the literature suggested by the reviewer. 

Point 2: Lines 73-100: This section looks like a methodological section. Thus, authors could add all of this explanation to materials and methods or rewrite them as an initial application of the methods used in this research, but all of them should be before Lines 62-72, since most research articles contain the objective and hypothesis at the end of the introduction. 

Response 2: We have now moved the hypothesis and the aim of the study at the end of the introduction. However, we feel that the data analytics (machine learning) tools used in this study do need to be introduced. Although we understand that the part in question makes the intro a bit technical, we believe that it serves the reader to get acquainted to the major machine learning technique used in our study and most importantly gives us the opportunity to explain the logic of the choices we made in performing certain type of analysis as we did. For instance why we performed a PCA and included the resulting components in the importance selection algorithm. We did try now to explain better certain steps.

Point 3: The authors have indicated that participants were apparently healthy. Did the authors conduct at least an overall analysis to confirm if they were healthy?

Response 3: This is a very valid point. We have indeed asked the participants to fill in a screening questionnaire. That was the American Heart Association/American College of Sports Medicine Pre-Participation
Questionnaire (AAPQ), which allowed us to exclude participants with cardiac diseases. However, it is true that we did not exclude participants with high glucose, lipids or insulin levels. We have added this extra-information in the text now. 

Point 4: Lines 107-109: Please provide the Ethics Committee Approval ID. 

Response 4: The EC Approval ID was actually already reported under the Institutional Review Board Statement.

Point 5: Lines 115-116: Please provide the full details from the ELISA test. This is important since detection limits vary highly between manufacturers. 

Response 5: Plasma glucose was analyzed by immobilized enzymatic assay (YSI 2300 STAT, Incorporated Life Sciences, Yellow Springs, OH, USA). Lipid profile was analyzed from plasma samples by optic enzymatic assay (ReXotron®, Roche Diagnostics, Mannheim, Germany). Plasma insulin was analyzed by ELISA (ultrasensitive human insulin ELISA kit, Mercodia, Uppsala, Sweden). Cytokines (IL-10, IL-6 and TNF-) and adipokines were also analyzed from fasting plasma samples by ELISA (Bender MedSystems GmbH, Vienna, Austria and BioVendor, Laboratoní medicína, Modrice, Czech Republic, respectively). Now added in the text.

Point 6: Line 239: Table 4 is mentioned here, but it is located very far away from this section. Authors should rearrange some figures or tables, so this table could be easily located. For instance, Fig. 2 and Fig. 3 could be merged in 1 figure. This is entirely a decision from the authors. 

Response 6: We appreciate the suggestion. We actually realized that table 4 was cited in Line 239 by mistake as PCA results were reported in the text and in the figures. Table 4 reports statistics of the mediation analysis. We have now rectified this error. About figure 2 and 3, figure 2 is still referring to PCA while figure 3 two the boruta importance selection, thus we think they should stay separate.

Round 2

Reviewer 1 Report

Thank you for addressing the issues.  The paper is clearer due to these changes.